

# Enstrophy from symmetry

**Raja Marjieh[1], Natalia Pinzani-Fokeeva[2,3,4*] and Amos Yarom[5]**

**1** Max Planck Institute for Empirical Aesthetics, Frankfurt am Main 60322, Germany
**2** Institute for Theoretical Physics, KU Leuven Celestijnenlaan 200D, Leuven B-3001, Belgium
**3** Dipartimento di Fisica e Astronomia, Universitá di Firenze,
Via G. Sansone 1, I-50019, Sesto Fiorentino (Firenze), Italy
**4** Center for Theoretical Physics, Massachusetts Institute of Technology,
Cambridge, MA 02139, USA
**5** Department of Physics, Technion, Haifa 32000, Israel

⋆ n.pinzanifokeeva@gmail.com

## Abstract

We study symmetry principles associated with the approximately conserved enstrophy current, responsible for the inverse energy cascade in non relativistic $2+1$ dimensional turbulence. We do so by identifying the accidental symmetry associated with enstrophy current conservation in a recently realized effective action principle for hydrodynamics. Our analysis deals with both relativistic and non relativistic effective actions and their associated symmetries.

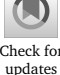

# 1   Introduction

Non relativistic incompressible fluid flow in two spatial dimensions supports an approximately conserved enstrophy charge whose existence plays a crucial role in generating the inverse energy cascade of turbulent flow [1]. Planar, non relativistic and compressible flow also supports an approximately conserved enstrophy charge as long as the equation of state is barotropic (the pressure is a function of the mass, or particle number density), see, e.g., [2]. Likewise, an approximately conserved enstrophy charge was shown to be present in relativistic and conformal invariant fluid flow in $2+1$ dimensions [3].

Charges which are conserved under the equations of motion are tied, via Noether's theorem, to symmetries of the underlying action. Thus, it stands to reason that there exists an approximate symmetry responsible for the approximate conservation of enstrophy. The goal of this work is to identify the symmetry associated with enstrophy conservation in relativistic and non relativistic fluid flows using a recently discovered action principle for fluids [4–6]. (See also [7–19].)

In local theories charge conservation follows from current conservation. It is interesting to contrast the approximate conservation of the enstrophy current with the behavior of the entropy current. Recall that the entropy current is given by $J_s^{\mu} = s u^{\mu} + \mathcal{O}(\partial)$ where $s$ is the entropy density, $u^{\mu}$ is the velocity field ($u^{\mu} = \gamma(1, v^i)$ in a relativistic setting and the same with $\gamma = 1$ in a non relativistic one), and $\mathcal{O}(\partial)$ denotes corrections which include derivatives of hydrodynamic variables. The entropy current is conserved in the absence of dissipative terms and its divergence is positive semi-definite otherwise. Thus, at leading order in a derivative expansion one may view the entropy current as being approximately conserved in the sense that $\partial_{\mu} J_s^{\mu} = \mathcal{O}(\partial^2)$ under the equations of motion. That is to say, the leading order contribution to the entropy current is of zeroth order in a derivative expansion, but, under the equations of motion, its divergence is second order. The approximate conservation of the enstrophy current is of a similar type. As we will see shortly, the enstrophy current is second order in derivatives but, under the equations of motion, its divergence is fourth order.

The analogy between the enstrophy current and the entropy current may run deeper than a comparison of their approximate conservation at leading order. In planar, non relativistic, incompressible or barotropic flow the enstrophy charge is conserved in the absence of dissipation but its time derivative is negative semi-definite once dissipation comes into play. This feature is crucial to the existence of an inverse energy cascade in 2+1 dimensions [1]. Whether such a behavior persists for relativistic fluid flows is yet an open problem. It is tentalizing to speculate that a negative semi-definite divergence of the enstrophy current may lead to an inverse energy cascade in relativistic fluids too.

Be that as it may, it is possible to identify the symmetry principle responsible for the conservation of the entropy current in the absence of dissipative terms using the hydrodynamic effective action [20–24]. In fact, one can identify the mechanism responsible for its full non conservation for generic, dissipative, fluids [9,14,15]. This raises the hope that a similar construction may be generated in order to better understand the enstrophy current. In this work we take a first step in this direction and find the symmetry associated with enstrophy conservation at leading order in the derivative expansion. Along the way we provide a rudimentary

construction of the effective action for Galilean fluids, offering a slightly different perspective on it than the recent comprehensive work of [25]. In addition, we identify an enstrophy current in generic relativistic fluids, generalizing the result of [3]. We will not discuss negativity of enstrophy production in a relativistic setting but will comment on its possible realization where relevant.

Our work is organized as follows. In Section 2 we discuss the structure of the relativistic and Galilean enstrophy currents. Our result for the relativistic enstrophy current generalizes that of [3], relevant for an uncharged conformal fluid, while our expression for the non relativistic enstrophy current has been cast in a manifestly covariant form by using the Newton-Cartan formalism. In Section 3 we discuss how a conserved enstrophy current arises as a Noether current of an effective action for fluid dynamics in 2+1 dimensions for both relativistic and Galilean fluid flows. We end with an outlook and discussion in Section 4. A review of the traditional approach to the construction of the enstrophy charge and a summary of Newton-Cartan geometry and its relation to Galilean invariant hydrodynamics have been relegated to the appendices.

## 2 The enstrophy current

In $2+1$ dimensional incompressible non relativistic fluid flow the enstrophy charge is given by

$$W = \frac{1}{2} \int \sqrt{g}\, \omega_{ij} \omega^{ij} d^2 x \,, \tag{1}$$

where $\omega_{ij}$ is the vorticity two form

$$\omega_{ij} = \partial_i v_j - \partial_j v_i \,, \tag{2}$$

with $v_i$ the velocity of a fluid element. The argument that $W$ is time independent in the absence of dissipation and negative semidefinite otherwise can be found in any textbook on hydrodynamics, e.g., [26]. We present the canonical derivation of this result in appendix A for completeness.

The total enstrophy $W$ in (1) may be interpreted as a volume integral over an enstrophy density which may be thought of as the zero component of an enstrophy current,

$$j^\mu_{(1)} = \omega_{ij} \omega^{ij} u^\mu_G \,, \tag{3}$$

with

$$u^\mu_G = (1, \vec{v}) \,. \tag{4}$$

The reason for the parenthetical (1) in (3) will become clear presently. The subscript $G$ in (4) stands for Galilean, to be distinguished from its Lorentzian counterpart which we will introduce shortly. The enstrophy current $j^\mu_{(1)}$ satisfies $\nabla_\mu j^\mu_{(1)} = 0$ at leading order in a derivative expansion and $\nabla_\mu j^\mu_{(1)} \leq 0$ otherwise.

To be somewhat more precise, there exists not one, but a family of enstrophy currents usually written in the form

$$j^\mu_{(n)} = (\omega_{ij} \omega^{ij})^n u^\mu_G \,, \tag{5}$$

with $n$ a positive integer. As was the case for $j^\mu_{(1)}$, $\nabla_\mu j^\mu_{(n)} = 0$ at leading order in the derivative expansion, and, as long as $n > 0$, $\nabla_\mu j^\mu_{(n)} \leq 0$ in the presence of dissipation. While non-standard, it is straightforward to argue that the currents (5) can be replaced with

$$j^\mu_h = h(\omega_{ij} \omega^{ij}) u^\mu_G \,, \tag{6}$$



where $h$ is a monotonically increasing function of its argument; one finds that $\nabla_\mu j_h^\mu \leq 0$ with a strict equality once the viscosity is set to zero.

Slightly less familiar is the family of conserved enstrophy currents associated with non relativistic, inviscid, compressible, and barotropic fluids,

$$j_G^\mu = \frac{g(s/\rho)}{s^{2n-1}} \left( \omega_{ij}\omega^{ij} \right)^n u_G^\mu. \tag{7}$$

Here, $\rho$ is the particle number density, $s$ the entropy density, and $g$ an arbitrary function. The barotropic condition states that $P = P(\rho)$ where $P$ is the pressure. Note that in incompressible flow the particle number density and the entropy density are constant so that (3) takes the same form as (7) in its regime of validity. The expression (7) can be replaced by

$$j_{Gh}^\mu = h(s/\rho, \omega_{ij}\omega^{ij}/s^2)su_G^\mu. \tag{8}$$

Following standard conventions we will, throughout this work, consider the version of the enstrophy current given in (7). Where relevant we will comment on the alternate form $j_{Gh}^\mu$ mentioned above.

An uncharged, inviscid, relativistic, conformal fluid in $2+1$ dimensions also possesses a conserved enstrophy current given by

$$J_{\text{conformal}}^\mu = \frac{\Omega^2}{\epsilon^{2/3}}u^\mu, \tag{9}$$

with $\epsilon$ the energy density, $u^\mu$ a relativistic velocity field and $\Omega^2 = \Omega_{\mu\nu}\Omega^{\mu\nu}$ where $\Omega_{\mu\nu} = \nabla_\mu\left(\epsilon^{1/3}u_\nu\right) - \nabla_\nu\left(\epsilon^{1/3}u_\mu\right)$. See [3]. In what follows we will generalize this result. In particular, we will argue that, in the presence of a $U(1)$ global symmetry, one can write a conserved enstrophy current for generic fluids of the form

$$J^\mu = \frac{g(s/\rho)}{s^{2n-1}}(\Omega^2)^n u^\mu, \tag{10}$$

with

$$\Omega_{\mu\nu} = \partial_\mu\left(Tf\left(\frac{\mu}{T}\right)u_\nu\right) - \partial_\nu\left(Tf\left(\frac{\mu}{T}\right)u_\mu\right). \tag{11}$$

Here, $T$ is the temperature, $\mu$ the chemical potential, $\rho$ a $U(1)$ charge density, and $f$ an arbitrary function of its argument. The current $J^\mu$ is conserved as long as the pressure, $P$, satisfies $P(T,\mu) = p(Tf(\mu/T))$. In the presence of an external electric field, $f$ becomes linear in $\mu$, and $\Omega_{\mu\nu}$ in (11) receives a contribution linear in the field strength, (see (26)). As was the case for the Galilean enstrophy current one may replace the family of conserved currents (10) with

$$J_h^\mu = h(s/\rho, \Omega^2/s^2)su^\mu. \tag{12}$$

## 2.1 The relativistic enstrophy current

Recall that the dynamical fields of hydrodynamic theory can be chosen to be the temperature $T$, a velocity field $u^\mu$ satisfying $u_\mu u^\mu = -1$, and a chemical potential $\mu$ if a conserved charge is present. The energy momentum tensor and other conserved currents of the theory can be expressed in terms of the dynamical fields and their derivatives. This description is usually made manifest in terms of a derivative expansion. For instance,

$$\begin{aligned} T^{\mu\nu} &= \epsilon(T,\mu)u^\mu u^\nu + P(T,\mu)(g^{\mu\nu} + u^\mu u^\nu) + \mathcal{O}(\partial), \\ J_c^\mu &= \rho(T,\mu)u^\mu + \mathcal{O}(\partial). \end{aligned} \tag{13}$$

Here, $\epsilon(T,\mu)$, $P(T,\mu)$ and $\rho(T,\mu)$ are functions of the temperature and chemical potential which, in equilibrium, reduce to the energy density, pressure and charge density respectively. The entropy density $s(T,\mu)$ and charge density $\rho(T,\mu)$ satisfy

$$s = \left(\frac{\partial P}{\partial T}\right)_\mu , \qquad \rho = \left(\frac{\partial P}{\partial \mu}\right)_T , \tag{14}$$

and

$$\epsilon + P = sT + \rho\mu . \tag{15}$$

In the inviscid limit, energy-momentum and charge current conservation, $\nabla_\mu T^{\mu\nu} = F^{\nu\mu}J_{c\,\mu}$ and $\nabla_\mu J_c^\mu = 0$, lead to the equations of motion $E = 0$, $E' = 0$, and $E_\mu = 0$ with

$$\begin{aligned} E &= -\nabla_\mu(su^\mu), \\ E_\mu &= \frac{(P+\epsilon)}{T}(P_\mu^\alpha \partial_\alpha T + Ta_\mu) - \rho V_\mu , \\ E' &= -\nabla_\mu(\rho u^\mu) . \end{aligned} \tag{16}$$

Here $a_\mu = u^\alpha \nabla_\alpha u_\mu$ is the acceleration, $P_{\alpha\beta} = g_{\alpha\beta} + u_\alpha u_\beta$ is a projection matrix and

$$V_\mu = F_{\mu\nu}u^\nu - TP_\mu^\nu \partial_\nu\left(\frac{\mu}{T}\right) , \tag{17}$$

with $F_{\mu\nu}$ an external field strength.

Suppose we find a closed two-form $\Omega_{\mu\nu}dx^\mu dx^\nu$ which is orthogonal to the velocity field, $\Omega_{\mu\nu}u^\nu = 0$, (at least under the equations of motion). Such a two-form satisfies

$$\mathcal{L}_u\Omega_{\mu\nu} = 0, \tag{18}$$

under the equations of motion, with $\mathcal{L}_u$ the Lie derivative in the $u^\mu$ direction. Using (18) and

$$\nabla_\mu u_\alpha = \frac{1}{2}\sigma_{\mu\alpha} + \frac{1}{2}\omega_{\mu\alpha} + \frac{1}{d}\Theta P_{\mu\alpha} - u_\mu a_\alpha , \tag{19}$$

where $d$ is the number of spatial dimensions and

$$\begin{aligned} \frac{1}{2}\sigma_{\mu\nu} &= \frac{1}{2}P_\mu{}^\alpha P_\nu{}^\beta\left(\nabla_\alpha u_\beta + \nabla_\beta u_\alpha\right) - \frac{1}{d}P_{\mu\nu}\nabla_\alpha u^\alpha , \\ \omega_{\mu\nu} &= P_\mu{}^\alpha P_\nu{}^\beta\left(\nabla_\alpha u_\beta - \nabla_\beta u_\alpha\right) , \\ \Theta &= \nabla_\alpha u^\alpha , \end{aligned} \tag{20}$$

we find that under the equations of motion

$$\Omega^{\mu\nu}\nabla_\alpha\left(u^\alpha\Omega_{\mu\nu}\right) = \Omega^{\mu\nu}\sigma_\nu{}^\alpha\Omega_{\alpha\mu} + \Theta\Omega^2\left(1 - \frac{2}{d}\right) . \tag{21}$$

In two spatial dimensions, the right-hand-side of (21) vanishes. That the first term is zero follows by denoting

$$\begin{aligned} \sigma_\nu{}^\alpha\Omega_{\alpha\mu} + \Omega_\nu{}^\alpha\sigma_{\alpha\mu} &= \epsilon_{\nu\mu\rho}u^\rho\sigma , \\ \Omega_{\mu\nu} &= \epsilon_{\mu\nu\rho}u^\rho\omega , \end{aligned} \tag{22}$$

where $\epsilon_{\mu\nu\rho}$ is the Levi-Civita tensor, and noting that

$$\sigma \propto \epsilon^{\mu\nu\rho}u_\rho\sigma_\nu{}^\alpha\epsilon_{\alpha\mu\lambda}u^\lambda\omega = 0 . \tag{23}$$

Thus, in two spatial dimensions, and after imposing the equations of motion,

$$\Omega^{\mu\nu}\nabla_\alpha\left(u^\alpha\Omega_{\mu\nu}\right)=0. \tag{24}$$

It is now straightforward to argue that $J^\mu$ given in (10) is conserved for any value of $n$ under the equations of motion. Note that if $n < 0$, then $J^\mu$ is ill defined in equilibrium. Also, $J^\mu = su^\mu$ coincides with the (inviscid) entropy current for $n = 0$ and $g = 1$. Likewise, $J^\mu = \rho u^\mu$ coincides with the charge current for $n = 0$ and $g = \rho/s$. The first term on the right-hand side of (21) bears a striking similarity to the vortex stretching term of non relativistic incompressible flow (see equation (98) in appendix A). Therefore, it is sensible to identify $J^\mu$ of (10) with $n = 1$ and $g = 1$ with the enstrophy current and $J^\mu$ with larger $n$ with its higher moments.

Using (24) one can also show that the current $J_h^\mu$ defined in (12) is also conserved under the equations of motion. It is also possible to generalize (10) to fluids with multiple $U(1)$ charges whereby $J^\mu = g\left(\frac{s}{\rho_1},\dots,\frac{s}{\rho_m}\right)\frac{(\Omega^2)^n}{s^{2n-1}}u^\mu$, with $\rho_i$ the various charge densities, is conserved. One might be tempted to construct an additional conserved current by contracting (24) with $\epsilon^{\mu\nu\rho}u_\rho$ to generate

$$J_H^\mu = H(s/\rho,\varpi/s)su^\mu, \tag{25a}$$

with

$$\varpi = u_\rho\epsilon^{\mu\nu\rho}\Omega_{\mu\nu}. \tag{25b}$$

It is straightforward to show that $\Omega^2 = \frac{1}{2}\varpi^2 + \mathcal{O}(E)$ implying that $J_H^\mu$ and $J_h^\mu$ are equivalent under the equations of motion.

The enstrophy current (10) was derived on the premise that a closed two-form $\Omega = \Omega_{\mu\nu}dx^\mu dx^\nu$, satisfying $\Omega_{\mu\nu}u^\nu = 0$ is available. To find it, let us start with the most general exact two-form which is first order in derivatives

$$\Omega_{\mu\nu} = \partial_\mu\left(Tf(T,\nu)u_\nu\right) - \partial_\nu\left(Tf(T,\nu)u_\mu\right) + \theta F_{\mu\nu}, \tag{26}$$

where $\theta$ is a constant, $\nu = \mu/T$ and $f$ is an arbitrary function of its variables. A somewhat lengthy computation yields

$$\Omega_{\mu\nu}u^\nu = -\frac{fT}{P+\epsilon}E_\mu + \left(\frac{f\rho T}{P+\epsilon} - \frac{\partial f}{\partial\nu}\right)TP_\mu^\alpha\partial_\alpha\nu - T\frac{\partial f}{\partial T}P_\mu^\alpha\partial_\alpha T - \left(\frac{f\rho T}{P+\epsilon} - \theta\right)F_{\mu\nu}u^\nu. \tag{27}$$

In order for the penultimate term on the right-hand side of (27) to vanish we need that

$$\frac{\partial f}{\partial T} = 0. \tag{28}$$

Solving for both (28) and the requirement that the second term on the right-hand-side of (27) vanish, we find that the equation of state must take the form

$$P(T,\mu) = p(Tf(\mu/T)). \tag{29}$$

Requiring that (27) vanishes under the equations of motion implies, in addition, that

$$f(\nu) = \theta\nu + \theta_0, \tag{30}$$

with $\theta_0$ an integration constant.

Let us summarize our findings. In the presence of an external electromagnetic field, a charged fluid must have an equation of state of the form (29) with (30) in order to possess a conserved enstrophy current. In the absence of an electromagnetic field, we must satisfy the somewhat less restrictive condition, (29), in order for $J^\mu$ of (10) to be conserved. Note that a charged conformal fluid and any neutral fluid will automatically have an equation of state of the form (29) and therefore possess a conserved enstrophy current (10).

## 2.2 The Galilean enstrophy current

The conserved enstrophy current for Galilean fluids can be constructed by borrowing the techniques used to construct the relativistic enstrophy current. A key feature of the construction of the relativistic enstrophy current was the existence of a closed two-form $\Omega_{\mu\nu}dx^\mu dx^\nu$ which was orthogonal to the velocity field under the equations of motion. With this two-form at hand, and the decomposition (19), we were lead to (21) from which enstrophy conservation in $2+1$ dimensions followed.

To construct such a Galilean invariant two-form, and consequently a conserved enstrophy current, we use the Newton-Cartan formalism which allows one to couple a Galilean invariant theory to a background spacetime in a covariant way. Galilean boost invariance is ensured by requiring a certain "Milne invariance" of the background geometry. We summarize this and other salient features of the Newton-Cartan geometry in appendix B.1. Briefly, Newton-Cartan geometry is characterized by a spatial metric $h^{\mu\nu}$, two timelike vectors $n_\mu$ and $\bar{n}^\mu$ such that $h^{\mu\nu}n_\mu = 0$ and $n_\mu\bar{n}^\mu = 1$, and a gauge field $A_\mu$. From these data one constructs an inverse metric $\bar{h}_{\mu\nu}$ and a projection $P^\mu{}_\nu$ via (108b). Fluid dynamics in a background Newton-Cartan geometry can be described by introducing a timelike Galilean velocity field $u_G^\mu$ which satisfies $u_G^\mu n_\mu = 1$. We briefly review hydrodynamics in a Newton-Cartan geometry in appendix B.2. The interested reader is referred to [27] for a detailed exposition.

In a generic Newton-Cartan background geometry the equivalent of the decomposition (19) is

$$\tilde{\nabla}_\mu u_G^\nu = \frac{1}{2}\sigma_\mu{}^\nu + \frac{1}{2}\omega_\mu{}^\nu + \frac{2}{d}\tilde{P}_\mu{}^\nu\Theta + n_\mu a^\nu, \tag{31}$$

with the combinations

$$
\begin{aligned}
\sigma_\mu{}^\nu &= \tilde{h}_{\mu\alpha}\tilde{P}_\beta^\nu\left(\tilde{\nabla}^\alpha u_G^\beta + \tilde{\nabla}^\beta u_G^\alpha\right) - \frac{2}{d}\tilde{P}_\mu{}^\nu\Theta, \\
\omega_\mu{}^\nu &= \tilde{h}_{\mu\alpha}\tilde{P}_\beta^\nu\left(\tilde{\nabla}^\alpha u_G^\beta - \tilde{\nabla}^\beta u_G^\alpha\right), \\
a^\nu &= u_G^\alpha\tilde{\nabla}_\alpha u_G^\nu, \\
\Theta &= \tilde{\nabla}_\mu u_G^\mu.
\end{aligned}
\tag{32}
$$

Here, $\tilde{h}_{\mu\nu}$ and $\tilde{P}^\mu{}_\nu$ are given in (120), $\tilde{\nabla}_\mu$ is the Milne invariant covariant derivative constructed in (121), $\tilde{\nabla}^\alpha = h^{\alpha\beta}\tilde{\nabla}_\beta$ and in obtaining (32) we made repeated use of $\tilde{\nabla}_\mu h^{\alpha\beta} = 0$ together with $n_\mu\tilde{\nabla}_\alpha u_G^\mu = 0$. The latter follows from the requirement that $\tilde{\nabla}_\mu n_\nu = 0$. It is important to keep in mind that in the Newton-Cartan formalism the Christoffel connection has torsion, see (121). Following [27], we have chosen it to be timelike.

Using the Cartan formula $\mathcal{L}_u\Omega_{\mu\nu} = 0$, we find that, under the equations of motion,

$$\Omega^{\alpha\beta}\tilde{\nabla}_\mu(u_G^\mu\Omega_{\alpha\beta}) = -\Omega^{\alpha\beta}\Omega_{\alpha\mu}\sigma_\beta^\mu + \Theta\Omega^2\left(1 - \frac{2}{d}\right) - 2\Omega^{\alpha\beta}\tilde{T}_{\alpha\mu}^\nu u_G^\mu\Omega_{\nu\beta}, \tag{33}$$

where $\tilde{T}_{\alpha\beta}^\mu$ is the torsion tensor, and we have defined

$$\Omega^{\mu\nu} = h^{\mu\alpha}h^{\nu\beta}\Omega_{\alpha\beta}, \qquad \Omega^2 = \Omega^{\mu\nu}\Omega_{\mu\nu}. \tag{34}$$

Since torsion is timelike, $\tilde{T}_{\alpha\beta}^\mu = -u_G^\mu F_{\alpha\beta}^{(n)}$, c.f., (121), the last term on the right-hand-side of (33) vanishes under the equations of motion,

$$-2\Omega^{\alpha\beta}\tilde{T}_{\alpha\mu}^\nu u_G^\mu\Omega_{\nu\beta} = 2\Omega^{\alpha\beta}F_{\alpha\mu}^{(n)}u_G^\mu\left(u_G^\nu\Omega_{\nu\beta}\right) = 0. \tag{35}$$

Thus, (33) reduces to

$$\Omega^{\alpha\beta}\tilde{\nabla}_\mu(u_G^\mu\Omega_{\alpha\beta}) = -\Omega^{\alpha\beta}\Omega_{\alpha\mu}\sigma^\mu_\beta + \Theta\Omega^2\left(1 - \frac{2}{d}\right). \tag{36}$$

Equation (36) is the Galilean equivalent of (21): the last term on its right clearly vanishes in $d = 2$ spatial dimensions. The first term on the right-hand-side of (36) is a vortex stretching term which, as we will now show, also vanishes in $d = 2$ spatial dimensions. Let us work in a coordinate system where, locally, $u_G^\mu = (1,0)$. In this coordinate system we have $\Omega_{\alpha\mu} = \delta^i_\alpha\delta^j_\mu\epsilon_{ij}\omega$ with $\epsilon_{ij}$ the Levi-Civita tensor and $\omega$ a real number. It is now straightforward to compute

$$\Omega^{\mu\alpha}\Omega_{\mu\beta} = \frac{1}{2}\Omega^2\tilde{P}^\alpha_\beta, \tag{37}$$

from which

$$\Omega^{\alpha\beta}\Omega_{\alpha\mu}\sigma^\mu_\beta = 0, \tag{38}$$

follows.

Using (36), we find that in 2+1 dimensions and under the equations of motion

$$(\tilde{\nabla}_\mu - \tilde{\mathcal{G}}_\mu)J_G^\mu = 0, \tag{39}$$

where $\tilde{\mathcal{G}}_\mu$ was defined in (126) and $J_G^\mu$ is given by

$$J_G^\mu = g\left(\frac{\rho}{s}\right)\frac{(\Omega^2)^n}{s^{2n-1}}u_G^\mu. \tag{40}$$

Equation (40) is a covariant version of (7). In obtaining (39) we repeatedly used the fact that $\tilde{\mathcal{G}}_\mu u_G^\mu = 0$ and $\tilde{\nabla}_\alpha h^{\mu\nu} = 0$.[1] As in the relativistic case, conservation of

$$J_{Gh}^\mu = h(s/\rho, \Omega^2/s^2)su_G^\mu, \tag{41}$$

also follows from (36). Moreover, contraction of (36) with $\epsilon^{\mu\nu\rho}n_\rho$ leads to a conserved $J_{GH}^\mu = H(s/\rho, \varpi/s)su_G^\mu$ where $\varpi = n_\rho\epsilon^{\mu\nu\rho}\Omega_{\mu\nu}$. Since $\Omega^2 = \frac{1}{2}\varpi^2 + \mathcal{O}(E)$, $J_{Gh}^\mu$ and $J_{GH}^\mu$ are equivalent up to terms proportional to the equations of motion.

It remains to find a closed and velocity orthogonal $\Omega_{\mu\nu}$. The most general $U(1)$ and Milne invariant closed two-form $\Omega_{\mu\nu}$ that can be constructed using the Newton-Cartan data is given by

$$\Omega_{\mu\nu} = \tilde{F}_{\mu\nu} + \partial_\mu(qn_\nu) - \partial_\nu(qn_\mu), \tag{42}$$

(up to a multiplicative constant which we set to 1 without loss of generality) where $q$ is a generic function of the entropy density, $s$, and particle number density, $\rho$, and $\tilde{F}_{\mu\nu}$ is the Milne invariant field strength defined in (116). Contracting one of the indices of (42) with the velocity field and using the equations of motion (127) we find

$$\Omega_{\mu\nu}u_G^\nu = \left(\frac{1}{\rho}\frac{\partial P}{\partial\rho} + \frac{\partial q}{\partial\rho}\right)\tilde{P}^\alpha_\mu\partial_\alpha\rho + \left(\frac{1}{\rho}\frac{\partial P}{\partial s} + \frac{\partial q}{\partial s}\right)\tilde{P}^\alpha_\mu\partial_\alpha s + \left(q + \frac{(P+\epsilon)}{\rho}\right)F_{\mu\nu}^{(n)}u_G^\nu. \tag{43}$$

In the absence of torsion, $F_{\mu\nu}^{(n)} = \partial_\mu n_\nu - \partial_\nu n_\mu = 0$, we find that the right-hand-side of (43) vanishes for an equation of state of the form

$$P = P(\rho), \tag{44}$$

---

[1] If the torsion tensor is not timelike then $\tilde{\mathcal{G}}_\mu u_G^\mu \neq 0$. Nevertheless, it is possible to show that the enstrophy current (40) is conserved, in the sense of (39), as long as a closed two-form $\Omega_{\mu\nu}dx^\mu dx^\nu$ orthogonal to the velocity field exists.

and

$$q = -\int \frac{1}{\frac{\partial P}{\partial \mu}} \frac{\partial P}{\partial \rho} d\rho = -\mu + c_0(T). \tag{45}$$

In the presence of torsion we need to require, in addition, that

$$\rho = \rho(\mu + c(T)), \tag{46}$$

(where $c_0(T) = -Tc'(T)$).

So far we have considered generic flows. In the context of fluid flow at low velocities it is also interesting to consider subsonic flow where the fluid becomes incompressible, see, e.g., [28]. In this limit the particle number density becomes constant, so that the equations of motion reduce to the incompressible Navier-Stokes equations. Put differently, incompressible flow can be thought of as a particular class of solutions to the equations of motion where the particle number density, and consequently the entropy density, are constant, and the pressure becomes an independent function of the coordinates. In torsionless, incompressible flow, equation (43) is automatically satisfied for an arbitrary choice of $q$. In the presence of torsion we must require $q = -(P + \epsilon)/\rho$.

To relate the covariant expressions (40), (42) and (44) to (5) and (7) we take the flat, torsionless, spacetime limit of (40) defined in (128). The enstrophy current $J_G^\mu$ in (40) reduces to (7). For incompressible flow, the particle number density, and therefore the entropy density become constant in which case (40) reduces to (5) up to an overall constant.

## 3 Enstrophy from symmetry

As stated in the introduction, it stands to reason that the enstrophy current of hydrodynamics is a result of an emergent symmetry of the theory. In what follows, we will use a recently developed formalism which allows one to construct an effective action for hydrodynamics [4–6] in order to identify the symmetry associated with enstrophy conservation. We will start with the relativistic enstrophy current for which the effective action has been studied in detail and then move on to the non relativistic theory where some extra ingredients are necessary in order to construct the effective action and derive the symmetry associated with (approximate) enstrophy conservation.

### 3.1 Relativistic enstrophy from symmetry

An effective action for an ideal charged fluid can be written in terms of a set of dynamical fields $X^\mu(\sigma)$ and $C(\sigma)$,

$$S_{eff}(X^\mu, C; \beta^i, \Lambda_\beta, g_{\mu\nu}, A_\mu) = \int \sqrt{-|g_{ij}|} P(T, \mu) d^{d+1}\sigma. \tag{47}$$

The function $X^\mu(\sigma)$ may be thought of as a dynamical Eulerian coordinate specifying the location of the fluid in a target space and $C(\sigma)$ an equivalent function specifying its phase under a global $U(1)$ symmetry. The parameters $\beta^i$ and $\Lambda_\beta$ specify the configuration of the fluid in the far past, and $g_{\mu\nu}$ and $A_\mu$ specify the metric and $U(1)$ flavor field of the target space where the fluid resides. The fields $g_{ij}$, $T$ and $\mu$ are defined via

$$g_{ij}(X) = \partial_i X^\mu \partial_j X^\nu g_{\mu\nu}(X), \qquad \beta^i g_{ij}\beta^j = -T^{-2}, \qquad \frac{\mu}{T} = \beta^i \left(\partial_i X^\mu A_\mu(X) + \partial_i C\right) + \Lambda_\beta, \tag{48}$$

and $P$ is a real function. By computing the stress tensor one finds that $P$ can be identified with the pressure, $T$ with the temperature and $\mu$ with the chemical potential. Other actions for

ideal or inviscid fluids can be found in [29–34]. We have chosen (47) since by doubling the fields (and adding appropriate ghosts) the action can be extended to include non dissipative fluids. We refer the reader to [35] for an extensive discussion.

We claim that the following transformation of the dynamical fields

$$\delta X^\mu = \frac{1}{Ts^2}\Omega^2 u^\mu - \frac{2}{sp'^2}E^\mu\Theta - \frac{4}{sp'}P^{\mu\beta}\Omega_{\beta\alpha}a^\alpha + \frac{4}{p'}P^{\mu\beta}\nabla_\alpha\left(\frac{1}{s}\Omega^\alpha{}_\beta\right),$$
$$\delta C = \frac{\mu}{Ts^2}\Omega^2 - A_\alpha\delta X^\alpha,$$

(49)

is a symmetry of the action in 2+1 dimensions. Here $u^\mu = \partial_i X^\mu \beta^i T$, and the remaining objects are related to $u^\mu$, $T$, $\mu$ and $P$ as in section 2. For instance, $p'$ is the derivative of $p$ with respect to its argument (see (29)). Further, the symmetry (49) leads to a conserved current

$$J'^\mu = \frac{\Omega^2}{s}u^\mu + \frac{4}{sp'}\Omega^{\mu\nu}E_\nu,$$

(50)

which we may identify with the enstrophy current (10) with $g = 1$ and $n = 1$ once the equations of motion are satisfied. We will generalize (49) and the associated (50) to obtain the class of currents (10) shortly.

To see that (49) is indeed a symmetry and leads to (50) let us consider a generic variation $\delta X^\mu$ and $\delta C$ of the action. The equations of motion for $\delta X^\mu$ are energy-momentum conservation in the target space and the equation of motion for $\delta C$ is current conservation. Thus,

$$\delta_X S_{eff} = -\int d^{d+1}\sigma\sqrt{-|g_{ij}|}\left(\left(\nabla_\mu T^\mu{}_\nu - F_\nu{}^\mu J_{c\mu} + A_\nu\nabla_\mu J_c^\mu\right)\delta X^\nu + \nabla_\mu J_c^\mu\delta C + \left(\begin{smallmatrix}\text{total}\\\text{derivative}\end{smallmatrix}\right)\right).$$

(51)

If the transformations $\delta X^\mu$ and $\delta C$ are a symmetry of the action, then $\delta_X S_{eff} = 0$ independently of the equations of motion. Therefore, if $\delta X^\mu$ and $\delta C$ are symmetries,

$$\left(\nabla_\mu T^\mu{}_\nu - F_\nu{}^\mu J_{c\mu} + A_\nu\nabla_\mu J_c^\mu\right)\delta X^\nu + \nabla_\mu J_c^\mu\delta C = \nabla_\mu S^\mu,$$

(52)

with $S^\mu$ a local current. The symmetries which will generate the enstrophy current should lead to $S^\mu = J^\mu$ up to possible extra terms proportional to the equations of motion. Using the expression for $J^\mu$ in (10) with $g = 1$ and $n = 1$ we find

$$\nabla_\mu J^\mu = \frac{1}{s^2}E\Omega^2 - \frac{2}{sp'^2}(E^\alpha E_\alpha)\Theta + \frac{4}{sp'}\Omega_{\alpha\beta}a^\alpha E^\beta + \frac{4}{p'}\nabla_\alpha\left(\frac{1}{s}\Omega^{\alpha\beta}\right)E_\beta - 4\nabla_\alpha\left(\frac{\Omega^{\alpha\beta}E_\beta}{p's}\right). \quad (53)$$

Inserting (53) into (52) and solving for $\delta X^\mu$ and $\delta C$ will give us transformations which can not be written in terms of positive powers of the equations of motion or their derivatives. To remedy this, we use $S^\mu = J'^\mu$ which leads to

$$(E_\alpha - TEu_\alpha - T\mu E'u_\alpha)\delta X^\alpha - E'(A_\alpha\delta X^\alpha + \delta C) =$$
$$+ \frac{1}{s^2}E\Omega^2 - \frac{2}{sp'^2}(E^\alpha E_\alpha)\Theta + \frac{4}{sp'}\Omega_{\alpha\beta}a^\alpha E^\beta + \frac{4}{p'}\nabla_\alpha\left(\frac{1}{s}\Omega^{\alpha\beta}\right)E_\beta.$$

(54)

(Note that covariance of (54) is ensured due to $\delta C \to -\delta X^\alpha\partial_\alpha\Lambda$ under gauge transformations.) One can check that the $\delta X^\mu$ and $\delta C$ given in (49) satisfy (54).

Symmetries associated with conserved currents constructed from higher powers of $\Omega^2$ as in (10), can be obtained in a similar fashion. Using $S^\mu = J^\mu + \mathcal{O}(E)$ in (52) we find that

$$
\delta X^\mu = \frac{(2n-1)g}{Ts^{2n}}(\Omega^2)^n u^\mu - \frac{g'}{Ts^{2n-1}\rho}(\Omega^2)^n u^\mu - \frac{2ng}{s^{2n-1}p'^2}(\Omega^2)^{n-1}E^\mu\Theta
$$
$$
- \frac{4ng}{s^{2n-1}p'}(\Omega^2)^{n-1}P^\mu{}_\beta\Omega^{\beta\alpha}a_\alpha + \frac{4n}{p'}P^\mu{}_\beta\nabla_\alpha\left(\frac{g}{s^{2n-1}}(\Omega^2)^{n-1}\Omega^{\alpha\beta}\right), \tag{55}
$$
$$
\delta C = \frac{(2n-1)\mu g}{Ts^{2n}}(\Omega^2)^n - \frac{\mu g'}{Ts^{2n-1}\rho}(\Omega^2)^n - \frac{g'}{s^{2(n-1)}\rho^2}(\Omega^2)^n - A_\alpha\delta X^\alpha,
$$

leads to the conserved current

$$
J'^\mu = g\left(\frac{s}{\rho}\right)\frac{(\Omega^2)^n}{s^{2n-1}}u^\mu + g\left(\frac{s}{\rho}\right)\frac{4n}{s^{2n-1}p'}(\Omega^2)^{n-1}\Omega^{\mu\nu}E_\nu. \tag{56}
$$

For $n = 1$ and $g = 1$ we recover (49) and (50) as expected. For completeness we note that

$$
\delta X^\mu = \frac{1}{T}\left(\frac{2\dot{h}}{s^2}\Omega^2 - \frac{s}{\rho}h' - h\right)u^\mu - \frac{4\dot{h}}{sp'}P^{\mu\alpha}\Omega_{\alpha\beta}a^\beta - \frac{2\dot{h}}{sp'^2}\Theta E^\mu + \frac{4}{p'}P^\mu{}_\beta\nabla_\alpha\left(\frac{\dot{h}}{s}\Omega^{\alpha\beta}\right),
$$
$$
\delta C = \frac{\mu}{T}\left(\frac{2\dot{h}}{s^2}\Omega^2 - \frac{s}{\rho}h' - h\right) - \frac{s^2}{\rho^2}h' - A_\alpha\delta X^\alpha, \tag{57}
$$

generate $J_h^\mu$ as defined in (12). In (57) we have defined $h'$ and $\dot{h}$ to be the derivatives with respect to the first and second argument of $h$ respectively.

The simplest symmetry that arises from (55) is given for $n = 0$ and $g = 1$. In that case,

$$
\delta X^\mu = -\frac{u^\mu}{T}, \qquad \delta C = -\frac{\mu}{T} - A_\alpha\delta X^\alpha, \tag{58}
$$

and the corresponding current is the entropy current $J_s = su^\mu$ as previously identified in [20–24]. Analogously, for $n = 0$ and $g = \rho/s$, we have

$$
\delta X^\mu = 0, \qquad \delta C = 1, \tag{59}
$$

which leads to conservation of the charge current $J = \rho u^\mu$. Unfortunately, neither (49) nor (55) nor (57) seem to provide a physically meaningful insight into the symmetry responsible for enstrophy conservation for $n \neq 0$. The simplest expression we were able to extract from (55) is

$$
\delta X^\alpha = \frac{1}{\sqrt{2}}\frac{4}{p'}\epsilon^{\alpha\beta\rho}u_\beta a_\rho, \qquad \delta C = -A_\alpha\delta X^\alpha, \tag{60}
$$

(up to the equations of motion), obtained by setting $n = 1/2$ and $g = 1$. The transformation (60) is associated with $J_H'^\mu = \frac{1}{\sqrt{2}}\left(\varpi u^\mu + \frac{2}{p'}\epsilon^{\mu\alpha\beta}E_\alpha u_\beta\right)$ obtained from (25) with $H = \frac{1}{\sqrt{2}}\varpi/s$. It is unclear whether the divergence of the latter current is sign definite.

So far we have worked with a Lagrange description of the fluid. It is possible to relate the symmetry (49) to a symmetry of the Eulerian degrees of freedom, $u^\mu$, $T$ and $\mu$. This symmetry can be easily found by considering the pushforwards of the initial state data through the dynamical degrees of freedom $X^\mu$ and $C$,

$$
\beta^\mu = \partial_i X^\mu \beta^i(\sigma(X)), \qquad \bar{\Lambda}_\beta = \Lambda_\beta(\sigma(X)) + \beta^\mu\partial_\mu C(\sigma(X)), \tag{61}
$$

which, under a change $\delta X^\mu$ and $\delta C$, transform as

$$
\delta\beta^\mu = -\mathcal{L}_{\delta X}\beta^\mu, \qquad \delta\bar{\Lambda}_\beta = -\mathcal{L}_{\delta X}\bar{\Lambda}_\beta + \beta^\mu\partial_\mu\delta C, \tag{62}
$$

where $\mathcal{L}_{\delta X}$ is the Lie derivative in the $\delta X^\mu$ direction with $\delta X^\mu$ and $\delta C$ defined in (49). Using (62) and the definitions of the Eulerian variables in the target space

$$T = \frac{1}{\sqrt{-\beta^\mu \beta^\nu g_{\mu\nu}}}, \qquad u^\mu = T\beta^\mu, \qquad \mu = u^\mu A_\mu + \bar{\Lambda}_\beta, \tag{63}$$

the symmetry (49), or more generally (55), acts on the conventional degrees of freedom as

$$
\begin{aligned}
\delta u^\mu &= -T P^\mu_\nu \mathcal{L}_{\delta X} \beta^\nu, \qquad \delta T = -T^2 u_\nu \mathcal{L}_{\delta X} \beta^\nu, \\
\delta \mu &= -\mu T u_\nu \mathcal{L}_{\delta X} \beta^\nu - T A_\nu \mathcal{L}_{\delta X} \beta^\nu - T \mathcal{L}_{\delta X} \bar{\Lambda}_\beta + u^\mu \partial_\mu \delta C,
\end{aligned}
\tag{64}
$$

while it is inert on the target space sources

$$\delta g_{\mu\nu} = 0, \qquad \delta A_\mu = 0. \tag{65}$$

## 3.2 Galilean enstrophy from symmetry

The procedure described in the previous section for obtaining the symmetry which generates the relativistic enstrophy current can be readily generalized to Galilean invariant systems. In what follows we first describe the ingredients required to construct an effective action for Galilean invariant fluids and then proceed to identify the symmetry associated with approximate conservation of enstrophy.

### 3.2.1 A Galilean effective action for hydrodynamics

A Schwinger-Keldysh effective action for Galilean fluids can be constructed from a higher dimensional relativistic one by equipping the latter with a null Killing vector [36]. This procedure was carried out in detail in [25]. Here, we will use an alternate construction similar to the one used to formulate the Schwinger-Keldysh effective action for relativistic fluids, or any infrared action for that matter. Namely, we identify the symmetries and dynamical fields associated with the fluid and then construct the most general action compatible with those symmetries. Since a full construction of the Schwinger-Keldysh effective action for Galilean fluids is available in [25] and since the various conceptual hurdles for constructing effective actions for fluids were described in detail in [4–19], we will be somewhat sparse in our exposition.

In a Newton-Cartan background geometry, the effective action should be invariant under coordinate reparameterizations, $x^\mu \to x^\mu + \xi^\mu$, the $U(1)$ gauge symmetry with parameter $\Lambda$, and Milne boosts with parameter $\psi_\nu$. When acting on the the Newton-Cartan data, these transformations take the form

$$
\begin{aligned}
\delta_\chi n_\mu &= \mathcal{L}_\xi n_\mu, \\
\delta_\chi h^{\mu\nu} &= \mathcal{L}_\xi h^{\mu\nu}, \\
\delta_\chi \bar{n}^\mu &= \mathcal{L}_\xi \bar{n}^\mu + h^{\mu\nu} \psi_\nu, \\
\delta_\chi A_\mu &= \mathcal{L}_\xi A_\mu + \partial_\mu \Lambda + P^\nu_\mu \psi_\nu - \frac{1}{2} n_\mu \psi^2, \\
\delta_\chi \bar{h}_{\mu\nu} &= \mathcal{L}_\xi \bar{h}_{\mu\nu} - \left( n_\mu P^\lambda_\nu + n_\nu P^\lambda_\mu \right) \psi_\lambda + n_\mu n_\nu \psi^2,
\end{aligned}
\tag{66}
$$

where $\psi^2 = \psi_\nu \psi_\rho h^{\nu\rho}$ and $\delta_\chi$ denotes a target space coordinate reparameterization, a $U(1)$ gauge transformation and a Milne transformation. The inverse metric $\bar{h}_{\mu\nu}$ is defined in (108b).

The dynamical fields of the Galilean invariant effective action for fluid dynamics are given by the coordinates $X^\mu(\sigma)$ and a phase $C(\sigma)$. As is the case for relativistic fluid dynamics, the $X^\mu$ fields parameterize worldlines of fluid elements. They provide a mapping between a parameter space specified by the coordinate $\sigma^i$ which we refer to as a worldvolume and the

space where the fluid elements live in, which we refer to as the target space. Similarly, $C(\sigma)$ is the field that captures the local phase of each fluid element. The astute reader will note that in addition to $X^\mu(\sigma)$ and $C(\sigma)$, one may have included a field $\phi_\mu(\sigma)$ which maps Milne transformations from the target space to the worldvolume. Worldvolume quantities which are not Milne invariant could then be rendered as such by modifying them with appropriate factors of $\phi_\mu(\sigma)$. As we shall see shortly all worldvolume quantities are explicitly Milne invariant so that $\phi_\mu(\sigma)$ will not appear in the effective action.

The dynamical variables are bifundamental fields and as such transform under the target space symmetries as well as under the corresponding symmetries induced on the worldvolume: worldvolume reparameterizations labeled by $\hat{\xi}^i$, worldvolume $U(1)$ gauge transformations with parameter $\hat{\Lambda}$, and worldvolume Milne boosts parameterized by $\hat{\psi}_i$,

$$
\begin{aligned}
\delta_{(\chi,\hat{\chi})}X^\mu(\sigma) &= -\xi^\mu(X(\sigma)) + \hat{\xi}^i(\sigma)\partial_i X^\mu(\sigma), \\
\delta_{(\chi,\hat{\chi})}C(\sigma) &= -\Lambda(X(\sigma)) + \hat{\Lambda}(\sigma) + \mathcal{L}_{\hat{\xi}}C(\sigma),
\end{aligned}
\tag{67}
$$

with $\delta_{\hat{\chi}}$ denoting worldvolume transformations. Had we added $\phi_\mu(\sigma)$, we would have found

$$
\delta_{(\chi,\hat{\chi})}\phi_\mu(\sigma) = -P_\mu^{\;\nu}(X(\sigma))\psi_\nu(X(\sigma)) + P_\mu^{\;\nu}(X(\sigma))(\partial_i X^\nu)^{-1}\hat{\psi}_i(\sigma) + \hat{\xi}^i(\sigma)\partial_i \phi_\mu(\sigma) - \mathcal{L}_\xi \phi_\mu(\sigma).
\tag{68}
$$

As should be clear from (67), $\delta X^\mu$ and $\delta C$ are both invariant under worldvolume Milne transformations.

In addition to the dynamical fields, the effective action will depend on the initial state data which specifies the equilibrium state of the system in the infinite past. This consists of a timelike Killing vector, $\beta^i(\sigma)$, specifying the initial velocity and temperature, a gauge Killing parameter, $\Lambda_\beta(\sigma)$, associated with the initial chemical potential, and a Milne boost one-form $\psi_\beta^i(\sigma)$. Since the system is in equilibrium in the infinite past the mapping between the target space and worldvolume is trivial. Thus,

$$
\begin{aligned}
\delta_\beta n_\mu(t=-\infty) &= 0, \quad \delta_\beta h^{\mu\nu}(t=-\infty) = 0, \\
\delta_\beta \bar{n}^\mu(t=-\infty) &= 0, \quad\;\; \delta_\beta A_\mu(t=-\infty) = 0,
\end{aligned}
\tag{69}
$$

where $\delta_\beta$ collectively denotes a worldvolume transformation given in (74) with parameters $\{\beta^i, \Lambda_\beta, \psi_i^\beta\}$. Worldvolume coordinate reparameterizations and $U(1)$ gauge transformations acting on the initial data take the form:[2]

$$
\begin{aligned}
\delta_{\hat{\chi}}\beta^i &= \mathcal{L}_{\hat{\xi}}\beta^i, \\
\delta_{\hat{\chi}}\Lambda_\beta &= \mathcal{L}_{\hat{\xi}}\Lambda_\beta - \beta^i \partial_i \hat{\Lambda}, \\
\delta_{\hat{\chi}}\psi_i^\beta &= \mathcal{L}_{\hat{\xi}}\psi_i^\beta - \mathcal{L}_\beta \hat{\psi}_i + \hat{\psi}_i.
\end{aligned}
\tag{70}
$$

Notice that $\beta^i$ and $\Lambda_\beta$ are invariant under Milne boosts while $\psi_i^\beta$ transforms non trivially under it.

The local effective action for Galilean fluids, $S_{eff}$, is constructed from worldvolume and target space invariant combinations of the dynamical fields and initial state data. In practice,

---

[2]Note that it is always possible to choose a gauge where the parameters specifying the initial data are fixed. A common choice is the static gauge where $\beta^i = b(1,\vec{0})$, with $b$ a constant, $\Lambda_\beta = 0$ and $\psi_i^\beta = 0$. As a result worldvolume transformations of the initial data will be restricted to a subset preserving the static gauge. We refrain from choosing a gauge in order to retain an explicitly covariant formulation of the action.

it is convenient to define the target space invariant quantities

$$
\begin{aligned}
n_i(\sigma) &= \partial_i X^\mu n_\mu(X), \\
h^{ij}(\sigma) &= (\partial_i X^\mu)^{-1}(\partial_j X^\nu)^{-1} h^{\mu\nu}(X), \\
\tilde{A}_i(\sigma) &= \partial_i X^\mu \tilde{A}_\mu(X) + \partial_i C, \\
\tilde{h}_{ij}(\sigma) &= \partial_i X^\mu \partial_j X^\nu \tilde{h}_{\mu\nu}(X),
\end{aligned}
\tag{71}
$$

where the tilde'd quantities

$$
\begin{aligned}
\tilde{A}_\mu &= A_\mu + u_{G\,\mu} - \frac{1}{2} n_\mu u_G^2, \\
\tilde{h}_{\mu\nu} &= \bar{h}_{\mu\nu} - u_{G\,\mu} n_\nu - u_{G\,\nu} n_\mu + n_\mu n_\nu u_G^2,
\end{aligned}
\tag{72}
$$

with

$$
u_G^\mu(X) = \frac{1}{\beta^i n_i} \beta^j \partial_j X^\mu, \qquad u_{G\,\mu} = \bar{h}_{\mu\nu} u_G^\nu, \qquad u_G^2 = u_{G\,\mu} u_G^\mu,
\tag{73}
$$

are Milne invariant. Note that had we not used the target space Milne invariant variables $\tilde{A}_\mu$ and $\tilde{h}_{\mu\nu}$ in (71), we would have been forced to use $\phi_\mu$ to ensure target space Milne invariance of $\tilde{A}_i$ and $\tilde{h}_{ij}$. It is the absence of $\phi_\mu$ on the right-hand-side of (71) that ensures that it does not appear in the effective action. It is straightforward to show that the target space invariant combinations (71) transform under worldvolume reparameterizations and $U(1)$ gauge transformations induced by the transformations of the dynamical fields (67) as

$$
\begin{aligned}
\delta_{\hat{\chi}} n_i &= \mathcal{L}_{\hat{\xi}} n_i, \\
\delta_{\hat{\chi}} h^{ij} &= \mathcal{L}_{\hat{\xi}} h^{ij}, \\
\delta_{\hat{\chi}} \tilde{A}_i &= \mathcal{L}_{\hat{\xi}} \tilde{A}_i + \partial_i \hat{\Lambda}, \\
\delta_{\hat{\chi}} \tilde{h}_{ij} &= \mathcal{L}_{\hat{\xi}} \tilde{h}_{ij},
\end{aligned}
\tag{74}
$$

and are invariant under worldvolume Milne boosts.

The symmetries on the worldvolume can be maintained by requiring the action to be a scalar that depends only on $U(1)$ gauge invariant and Milne invariant quantities. At leading order in derivatives, the unique scalar invariants are

$$
T = \frac{1}{\beta^i n_i}, \qquad \mu = T\beta^i \tilde{A}_i + T\Lambda_\beta,
\tag{75}
$$

corresponding respectively to the temperature and the chemical potential. Keeping all the symmetries intact, we find that the most general effective action for Galilean fluids at leading order in derivatives is

$$
S_{eff} = \int d^{d+1}\sigma \sqrt{\gamma}\, P(T, \mu),
\tag{76}
$$

where the measure is given by the (Milne invariant) determinant of $\gamma_{ij} = \partial_i X^\mu \partial_j X^\nu \bar{h}_{\mu\nu} + n_i n_j$ and $P$ is a generic function of the temperature $T$ and chemical potential $\mu$.

To get a feel for this formulation of Galilean hydrodynamics let us derive the equations of motion for (ideal) Galilean fluids by varying the effective action with respect to the dynamical variables. A generic variation of the effective action (76) is given by

$$
\delta S_{eff} = \int d^{d+1}\sigma \sqrt{\gamma} \left( P \frac{1}{\sqrt{\gamma}} \delta \sqrt{\gamma} + s\, \delta T + \rho\, \delta\mu \right),
\tag{77}
$$

where we have defined

$$s = \left(\frac{\partial P}{\partial T}\right)_{\mu} \qquad \text{and} \qquad \rho = \left(\frac{\partial P}{\partial \mu}\right)_{T}. \tag{78}$$

In order to write the variations specified in (77) in terms of variations of the dynamical variables, we first use (75) to write

$$
\begin{aligned}
\delta T &= -T u_G^i \delta n_i \,, \\
\delta \mu &= u_G^i \delta \tilde{A}_i - \mu u_G^i \delta n_i \,, \\
\frac{1}{\sqrt{\gamma}} \delta \sqrt{\gamma} &= \frac{1}{\sqrt{\tilde{\gamma}}} \delta \sqrt{\tilde{\gamma}} = u_G^i \delta n_i + \frac{1}{2} \tilde{\gamma}^{ij} \delta \tilde{h}_{ij} \,,
\end{aligned}
\tag{79}
$$

where we have defined $u_G^i = T \beta^i$ and

$$\tilde{\gamma}_{ij} = n_i n_j + \tilde{h}_{ij} \,, \qquad \tilde{\gamma}^{ij} = u_G^i u_G^j + h^{ij} \,. \tag{80}$$

To derive the last expression in (79) we have used

$$\delta \tilde{h}_{ij} = \tilde{P}_k^i \tilde{P}_l^j \delta \bar{h}_{kl} - (u_{Gi} - n_i u_G^2)\tilde{P}_j^k \delta n_k - (u_{Gj} - n_j u_G^2)\tilde{P}_i^k \delta n_k \,, \qquad \delta u_G^i = -u_G^i u_G^k \delta n_k \,, \tag{81}$$

with $u_G^2 = u_{Gi} u_G^i$ and $\tilde{P}_j^i = h^{ik}\tilde{h}_{kj} = \delta_j^i - u_G^i n_j$. In writing the generic variations in (79) we have not included variations with respect to the initial state data $\beta^i$ and $\Lambda_\beta$ since they do not depend on the dynamical variables.

Next, consider

$$
\begin{aligned}
\delta n_i &= \partial_i X^\mu \mathcal{L}_{\delta X} n_\mu \,, \\
\delta \tilde{A}_i &= \partial_i X^\mu \mathcal{L}_{\delta X} \tilde{A}_\mu + \partial_i \delta C \,, \\
\delta \tilde{h}_{ij} &= \partial_i X^\mu \partial_j X^\nu \tilde{h}_{\mu\nu} \,,
\end{aligned}
\tag{82}
$$

where $\mathcal{L}_{\delta X}$ is the Lie derivative along $\delta X^\mu$. Inserting (82) into (79) and then into (77) we find

$$\delta S_{eff} = -\int d^{d+1}\sigma \sqrt{\gamma} \left( \left(E_\mu + (TE + \mu E') n_\rho\right) \delta X^\rho - E'\left(\delta C + \tilde{A}_\rho \delta X^\rho\right) \right) \tag{83}$$

up to total derivatives. In writing (83) we have repeatedly used the relation

$$\frac{1}{\sqrt{\gamma}} \partial_\mu (\sqrt{\gamma} V^\mu) = (\nabla_\mu - \mathcal{G}_\mu) V^\mu = (\tilde{\nabla}_\mu - \tilde{\mathcal{G}}_\mu) V^\mu \,, \tag{84}$$

with $\tilde{\mathcal{G}}_\mu$ defined in (126). Satisfyingly, the expressions for $E$, $E'$ and $E_\mu$ coincide with those in (127). We reproduce them here for convenience,

$$
\begin{aligned}
E_\mu &= \tilde{P}_\mu^\alpha \partial_\alpha P - \rho \tilde{F}_{\mu\alpha} u_G^\alpha + (P + \epsilon) F^{(n)}_{\mu\alpha} u_G^\alpha \,, \\
E &= -(\tilde{\nabla}_\mu - \tilde{\mathcal{G}}_\mu)(s u_G^\mu) \,, \\
E' &= -(\tilde{\nabla}_\mu - \tilde{\mathcal{G}}_\mu)(\rho u_G^\mu) \,.
\end{aligned}
$$

### 3.2.2 Extracting the Galilean enstrophy from symmetry

The transformations of $\delta X^\mu$ and $\delta C$ which generate the symmetry associated with enstrophy conservation must satisfy

$$(E_\mu + (TE + \mu E') n_\mu)\delta X^\mu - E'(\delta C + \tilde{A}_\rho \delta X^\rho) = (\tilde{\nabla}_\mu - \tilde{\mathcal{G}}_\mu) S^\mu \,, \tag{85}$$

with

$$S^\mu = J_G^\mu + \mathcal{O}(E). \tag{86}$$

In 2+1 dimensions the expression in (33) reduces to

$$\Omega^{\alpha\beta}\tilde{\nabla}_\mu(u_G^\mu\Omega_{\alpha\beta}) = -2\Omega^{\mu\nu}\tilde{\nabla}_\mu\left(\frac{1}{\rho}E_\nu\right) + 2\Omega^{\mu\nu}F_{\mu\rho}^{(n)}u^\rho\left(\frac{1}{\rho}E_\nu\right), \tag{87}$$

and conservation of the enstrophy current defined in (40) reads

$$
\begin{aligned}
(\tilde{\nabla}_\mu - \tilde{\mathcal{G}}_\mu)J_G^\mu ={}& \frac{(2n-1)g}{s^{2n}}(\Omega^2)^n E + \frac{g'}{\rho\, s^{2n-1}}\left(\frac{s}{\rho}E' - E\right)(\Omega^2)^n \\
& - \frac{4ng}{s^{2n-1}}(\Omega^2)^{n-1}\Omega^{\alpha\beta}\tilde{\nabla}_\alpha\left(\frac{1}{\rho}E_\beta\right) + \frac{4ng}{s^{2n-1}\rho}(\Omega^2)^{n-1}\Omega^{\alpha\beta}F_{\alpha\mu}^{(n)}u_G^\mu E_\beta.
\end{aligned}
\tag{88}
$$

Defining

$$J_G'^\mu = J_G^\mu + \frac{4ng}{s^{2n-1}\rho}(\Omega^2)^{n-1}\Omega^{\mu\beta}E_\beta, \tag{89}$$

we find

$$
\begin{aligned}
(\tilde{\nabla}_\mu - \tilde{\mathcal{G}}_\mu)J_G'^\mu ={}& \frac{(2n-1)g}{s^{2n}}(\Omega^2)^n E + \frac{g'}{\rho\, s^{2n-1}}\left(\frac{s}{\rho}E' - E\right)(\Omega^2)^n \\
& + \frac{1}{\rho}E_\beta\tilde{\nabla}_\alpha\left(\frac{4ng}{s^{2n-1}}(\Omega^2)^{n-1}\Omega^{\alpha\beta}\right) + \frac{4ng}{s^{2n-1}\rho}(\Omega^2)^{n-1}\Omega^{\alpha\beta}F_{\alpha\mu}^{(n)}u_G^\mu E_\beta.
\end{aligned}
\tag{90}
$$

It is now straightforward to show that

$$
\begin{aligned}
\delta X^\mu ={}& \frac{1}{T}\frac{(2n-1)g}{s^{2n}}(\Omega^2)^n u_G^\mu - \frac{g'}{T\rho\, s^{2n-1}}(\Omega^2)^n u_G^\mu \\
& + \frac{1}{\rho}\tilde{P}_\beta^\mu\tilde{\nabla}_\alpha\left(\frac{4ng}{s^{2n-1}}(\Omega^2)^{n-1}\Omega^{\alpha\beta}\right) + \frac{4ng}{s^{2n-1}\rho}(\Omega^2)^{n-1}\Omega^{\alpha\mu}F_{\alpha\nu}^{(n)}u_G^\nu,
\end{aligned}
\tag{91}
$$

$$\delta C = \frac{\mu}{T}\frac{(2n-1)g}{s^{2n}}(\Omega^2)^n - \frac{\mu}{T}\frac{g'}{\rho\, s^{2n-1}}(\Omega^2)^n - \frac{g'}{\rho^2\, s^{2n-2}}(\Omega^2)^n - \tilde{A}_\rho\,\delta X^\rho,$$

satisfy the condition (85) with $S^\mu$ given by $J_G'^\mu$, defined in (89). For completeness we note that

$$\delta X^\mu = \frac{1}{T}\left(\frac{2\dot{h}}{s^2}\Omega^2 - \frac{s}{\rho}h' - h\right)u_G^\mu + \frac{4}{\rho}\tilde{P}^\mu{}_\beta\tilde{\nabla}_\alpha\left(\frac{\dot{h}}{s}\Omega^{\alpha\beta}\right) + \frac{4\dot{h}}{s\rho}\tilde{P}^\mu{}_\beta\Omega^{\alpha\beta}F_{\alpha\nu}^{(n)}u_G^\nu,$$

$$\delta C = \frac{\mu}{T}\left(\frac{2\dot{h}}{s^2}\Omega^2 - \frac{s}{\rho}h' - h\right) - \frac{s^2}{\rho^2}h' - \tilde{A}_\alpha\,\delta X^\alpha, \tag{92}$$

lead to the conservation of $J_{Gh}^\mu$ defined in (41).

## 4 Conclusions

In this work we used the recently discovered effective action for hydrodynamics to determine the approximate symmetry responsible for the approximately conserved enstrophy current in $2+1$ dimensional relativistic and Galilean flow. In the process of our analysis, we have identified a mechanism which allows for the construction of the enstrophy current and used it to generalize previously known results regarding its form.

The mechanism we identified for constructing the enstrophy current relies on the existence of a closed two-form $\Omega_{\mu\nu}dx^\mu dx^\nu$ orthogonal to the velocity field, $\Omega_{\mu\nu}u^\nu = 0$, at least under

the equations of motion. Once such a two-form is available the existence of the enstrophy current is guaranteed. We believe that this mechanism can be used to construct an enstrophy current for fluid flows which are not relativistic or Galilean. Fluid dynamics in the absence of boost invariance has been studied recently in [37–39] and may be relevant to a variety of physical systems, see, e.g., [40].

Our current analysis neglected dissipation, which, in the Galilean case, leads to a non trivial but sign definite change in enstrophy charge over time. This fact, together with conservation of energy, is a key ingredient in the argument leading to the inverse energy cascade in turbulent flow (see Appendix A). It is not known whether a relativistic enstrophy current whose divergence is sign (semi-)definite exists. In order to study this problem one would start with $J^\mu$ in (10) (setting, say, $n = 1$) and consider $\mathcal{O}(\partial^3)$ corrections to it such that its divergence is sign (semi-)definite up to $\mathcal{O}(\partial^4)$. The existence of a relativistic enstrophy current with a sign (semi-)definite divergence may have implications for relativistic turbulence in $2+1$ dimensions.

In the context of holography, the existence of an enstrophy current for conformal $2 + 1$ dimensional fluid flow implies its dual manifestation in asymptotically $AdS_4$ black brane geometries. More precisely, as is the case for entropy, one may expect that asymptotically $AdS_4$ black branes possess a geometric quantity that captures enstrophy conservation in the boundary theory. There are several approaches to this problem in the literature [41–43] which may serve as an excellent starting point for fully addressing this issue. Understanding the role of approximate entrophy conservation in asymptotically $AdS_4$ black branes may lead to novel insights in holographic turbulence. Even more relevant would be to understand whether an approximate enstrophy conservation law arises regardless of the holographic duality.

## Acknowledgments

We would like to thank A. Frishman and H. Liu for useful discussions. NPF is supported by the Bijzonder Onderzoeksfonds 2020 at KU Leuven and by the European Commission through the Marie Sklodowska-Curie Action UniCHydro (grant agreement ID: 886540). AY is supported in part by an Israeli Science Foundation excellence center grant 2289/18 and a Binational Science Foundation grant 2016324.

## A  The non relativistic enstrophy charge

As discussed in the main text it is straightforward to argue that the enstrophy is conserved in inviscid $2 + 1$ dimensional incompressible flow. Consider the Navier-Stokes equation for incompressible fluids in the absence of random forces

$$\partial_t \vec{v} + \vec{v} \cdot \vec{\nabla} \vec{v} + \vec{\nabla} P = \frac{1}{R} \nabla^2 \vec{v},$$
$$\vec{\nabla} \cdot \vec{v} = 0,$$
(93)

where $R$ is the Reynolds number, $P$ is the pressure, and $\vec{v}$ is the velocity field. We start by making two observations. By dotting the Navier Stokes equation into $\vec{v}$ we find that

$$\frac{1}{2}\partial_t v^2 + \frac{1}{2}\vec{\nabla} \cdot \left(\vec{v} v^2\right) + \vec{\nabla}\left(\vec{v} P\right) = \frac{1}{R}\left(-\frac{1}{2}\omega_{ij}\omega^{ij} + \nabla_j\left(v_i\nabla^j v^i - v_i\nabla^i v^j\right)\right),$$
(94)

where

$$v^2 = \vec{v} \cdot \vec{v}, \qquad \omega_{ij} = \partial_i v_j - \partial_j v_i,$$
(95)

and we have used the incompressibility condition. Integrating (94) we find

$$\partial_t E = -\frac{1}{R} W \,, \tag{96}$$

where

$$E = \frac{1}{2} \int \sqrt{g}\, v^2 d^d x \,, \qquad \text{and} \qquad W = \frac{1}{2} \int \sqrt{g}\, \omega_{ij} \omega^{ij} d^d x \,, \tag{97}$$

are referred to as the total energy and the total enstrophy respectively. In obtaining (96) we have assumed that the fluid is on a manifold without a boundary. We will not consider manifolds with boundaries in the remainder of this work. When $R^{-1} = 0$ then, unsurprisingly, energy is conserved.

To understand the role of enstrophy in establishing the dynamics of the theory, let us consider the equation of motion for $\omega_{ij}$. By taking a derivative of (93) we obtain

$$\partial_t \omega_{ij} + \nabla_k \left( v^k \omega_{ij} \right) + \frac{1}{2} \left( \omega_{ik} \sigma_j^k + \sigma_{ik} \omega_j^k \right) = \frac{1}{R} \nabla^2 \omega_{ij} \,, \tag{98}$$

where

$$\sigma_{ij} = \nabla_i v_j + \nabla_j v_i \,. \tag{99}$$

The third term from the left is referred to as a 'vortex stretching' term and it vanishes in 2 spatial dimensions. Indeed, let

$$\omega_{ik} \sigma_j^k + \sigma_{ik} \omega_j^k = \epsilon_{ij} s \,, \tag{100}$$

and also

$$\omega_{ij} = \epsilon_{ij} \omega \,. \tag{101}$$

Then,

$$s \propto \epsilon^{ij} \omega_{ik} \sigma_j^k = \omega \epsilon^{ij} \epsilon_{ik} \sigma_j^k = \sigma_j^j = 0 \,, \tag{102}$$

where the last equality follows from the incompressibility condition. The enstrophy production equation reads

$$\partial_t W = \int \sqrt{g}\, \omega_{ji} \omega^i{}_k \sigma^{kj} d^d x - \frac{1}{R} P \,, \tag{103}$$

where $P$ is the Palinstrophy,

$$P = \int \sqrt{g}\, \nabla_k \omega_{ij} \nabla^k \omega^{ij} d^d x \,. \tag{104}$$

In the presence of the vortex stretching term the rate of change of $W$ is not sign definite. In this case experimental results and indirect theoretical arguments lead to

$$\lim_{R^{-1} \to 0} \frac{W}{R} = e_0 \,, \tag{105}$$

where $e_0$ is a constant. With some work, (see, e.g., [44]) one can show that (105) leads to the Kolmogorov energy cascade in turbulent flow. Once the vortex stretching term is absent, it is easy to show that $\partial_t W \leq 0$. Since the enstrophy is a positive quantity, it can not diverge if it were initially finite and (105) is no longer valid. Instead one finds, via (96), that energy will be conserved at large Reynolds number leading, eventually, to an inverse energy cascade (and also a direct enstrophy cascade) in two dimensional turbulent flow.

We also note in passing that higher moments of the enstrophy are also monotonically decreasing and conserved when $R^{-1} = 0$, viz.

$$\partial_t \int \sqrt{g} \left(\omega_{ij}\omega^{ij}\right)^n d^2x = -\frac{n}{R} \int \sqrt{g} \left(\omega_{ij}\omega^{ij}\right)^{n-1} \nabla_k \omega_{ij} \nabla^k \omega^{ij} d^2x \,, \tag{106}$$

whenever $n > 0$. Alternately,

$$\partial_t \int \sqrt{g} \, h\left(\omega_{ij}\omega^{ij}\right) d^2x = -\frac{1}{R} \int \sqrt{g} \, h'\left(\omega_{ij}\omega^{ij}\right) \nabla_k \omega_{ij} \nabla^k \omega^{ij} d^2x \,, \tag{107}$$

is negative as long as $h$ is a monotonically increasing function.

## B  Newton-Cartan geometry and hydrodynamics

Galilean invariant dynamics in a curved background, and Galilean invariant hydrodynamics in particular, is properly described by Newton-Cartan geometry. In what follows we will briefly summarize key elements of the Newton-Cartan formalism developed in [45] and then use it to recast Galilean hydrodynamics in a manifestly covariant form. See [27].

### B.1  Newton-Cartan geometry

In $d+1$ spacetime dimensions, the independent Newton-Cartan background data can be taken to be the set $(n_\mu, h^{\mu\nu}, A_\mu, \bar{n}^\mu)$, where $n_\mu$ is a nowhere vanishing one-form which defines the local time direction, $h^{\mu\nu}$ is a rank $d$ positive semi-definite symmetric tensor which satisfies $h^{\mu\nu}n_\mu = 0$ and can be seen as defining the (inverse) spatial metric, $A_\mu$ is a $U(1)$ gauge field associated with the conservation of particle number, and $\bar{n}^\mu$ is related to $n_\mu$ via

$$\bar{n}^\mu n_\mu = 1 \,. \tag{108a}$$

Based on the Newton-Cartan data, one can define a positive-definite spacetime metric $\gamma^{\mu\nu}$ (and its inverse $\gamma_{\mu\nu}$), a rank $d$ (spatial) metric $\bar{h}_{\mu\nu}$ and a projector $P^\mu{}_\nu$ via

$$\gamma^{\mu\nu} = \bar{n}^\mu \bar{n}^\nu + h^{\mu\nu} \,, \qquad \bar{h}_{\mu\nu} = \gamma_{\mu\nu} - n_\mu n_\nu \,, \qquad P^\mu{}_\nu = h^{\mu\rho}\bar{h}_{\nu\rho} = \delta^\mu_\nu - \bar{n}^\mu n_\nu \,. \tag{108b}$$

Note that $\bar{h}_{\mu\nu}\bar{n}^\nu = 0$ and $P^\mu{}_\nu \bar{n}^\nu = P^\mu{}_\nu n_\mu = 0$.

In Newton-Cartan theory different choices of $\bar{n}^\mu$ are equivalent. This is a result of the requirement that the underlying theory is Galilean invariant. In practice, we require that the action is invariant under a transformation $\bar{n}^\mu \to \bar{n}'^\mu$ obtained via

$$\bar{n}'^\mu = \bar{n}^\mu + h^{\mu\nu}\psi_\nu \,, \tag{109}$$

with $\psi_\nu$ a transverse one-form, $\psi_\nu \bar{n}^\nu = 0$. The transformation (109) is referred to as a Milne boost. The action of Milne boosts on the metric and gauge field is given by

$$\begin{aligned}
\bar{h}'_{\mu\nu} &= \bar{h}_{\mu\nu} - (n_\mu P^\rho{}_\nu + n_\nu P^\rho{}_\mu)\psi_\rho + n_\mu n_\nu h^{\alpha\beta}\psi_\alpha\psi_\beta \,, \\
A'_\mu &= A_\mu + P^\nu{}_\mu \psi_\nu - \frac{1}{2} n_\mu h^{\nu\rho}\psi_\nu\psi_\rho \,,
\end{aligned} \tag{110}$$

with $h^{\mu\nu}$ and $n_\mu$ invariant. Invariance under Milne transformations is a key requirement used to construct Galilean invariant theories in a curved background.

The covariant measure appearing in spacetime integrals is $d^{d+1}x\,\sqrt{\gamma}$, where $\gamma = \det(\gamma_{\mu\nu})$, which can be shown to be Milne invariant. The derivative that reduces to a boundary term under the integral is given by the combination

$$(\nabla_\mu - \mathcal{G}_\mu)V^\mu = \frac{1}{\sqrt{\gamma}}\partial_\mu(\sqrt{\gamma}\,V^\mu)\,, \tag{111}$$

where we have defined

$$\mathcal{G}_\mu = T^\alpha_{\mu\alpha} = -F^{(n)}_{\mu\alpha}\bar{n}^\alpha\,, \tag{112}$$

with

$$F^{(n)}_{\mu\nu} = \partial_\mu n_\nu - \partial_\nu n_\mu\,. \tag{113}$$

A covariant derivative $\nabla_\mu$ can be constructed by requiring compatibility with the Newton-Cartan data,

$$\nabla_\mu n_\nu = 0\,, \qquad \nabla_\mu h^{\alpha\beta} = 0\,, \tag{114}$$

and restricting the torsion to be timelike, $\bar{h}_{\lambda\rho}T^\lambda_{\mu\nu} = 0$,

$$
\begin{aligned}
\Gamma^\lambda_{\mu\nu} &= \bar{n}^\lambda\partial_\nu n_\mu + \frac{1}{2}h^{\lambda\rho}\left(\partial_\mu\bar{h}_{\nu\rho} + \partial_\nu\bar{h}_{\mu\rho} - \partial_\rho\bar{h}_{\mu\nu}\right) + \frac{1}{2}h^{\lambda\rho}\left(n_\mu F_{\nu\rho} + n_\nu F_{\mu\rho}\right)\,, \\
T^\lambda_{\mu\nu} &= \Gamma^\lambda_{\mu\nu} - \Gamma^\lambda_{\nu\mu} = -\bar{n}^\lambda F^{(n)}_{\mu\nu}\,,
\end{aligned}
\tag{115}
$$

where we have defined the field strength

$$F_{\mu\nu} = \partial_\mu A_\nu - \partial_\nu A_\mu\,, \tag{116}$$

and we used the conventions

$$\nabla_\mu V^\alpha_\beta = \partial_\mu V^\alpha_\beta + \Gamma^\alpha_{\rho\mu}V^\rho_\beta - \Gamma^\rho_{\beta\mu}V^\alpha_\rho\,. \tag{117}$$

Our construction closely follows that of [45]. A more general analysis can be found in [46]. The connection (115) is not Milne invariant. Unfortunately, using only the Newton-Cartan data it is not possible to construct a connection that is both Milne invariant and gauge invariant, see [45]. As we will see shortly, when discussing the hydrodynamic theory, one can use the velocity field as additional data in order to construct Milne and $U(1)$ invariant connections, see [27].

## B.2  Fluids on a Newton-Cartan background

Let us now consider a fluid in a curved Newton-Cartan background geometry. Following [27], we equip our theory with a Milne-invariant timelike velocity vector field $u^\mu_G$ normalized such that $u^\mu_G n_\mu = 1$. We also define the lower index counterpart of $u^\mu_G$ and its norm as

$$u_{G\mu} = \bar{h}_{\mu\nu}u^\nu_G\,, \qquad u^2_G = u_{G\mu}u^\mu_G\,, \tag{118}$$

which transform under Milne boosts as

$$
\begin{aligned}
u'_{G\mu} &= u_{G\mu} - P^\nu_\mu\psi_\nu + n_\mu h^{\nu\rho}(\psi_\nu\psi_\rho - u_{G\nu}\psi_\rho)\,, \\
(u'_G)^2 &= u^2_G + h^{\mu\nu}\psi_\mu\psi_\nu - 2h^{\mu\nu}u_{G\mu}\psi_\nu\,.
\end{aligned}
\tag{119}
$$

With these quantities at hand, it is straightforward to construct the Milne invariant combinations

$$
\begin{aligned}
\tilde{h}_{\mu\nu} &= \bar{h}_{\mu\nu} - (u_{G\mu}n_\nu + u_{G\nu}n_\mu) + u^2_G n_\mu n_\nu\,, \\
\tilde{A}_\mu &= A_\mu + u_{G\mu} - \frac{1}{2}n_\mu u^2_G\,, \\
\tilde{P}^\mu{}_\nu &= h^{\mu\rho}\tilde{h}_{\rho\nu} = \delta^\mu{}_\nu - u^\mu_G n_\nu\,,
\end{aligned}
\tag{120}
$$

which satisfy $\tilde{h}_{\mu\nu}u_G^\nu = 0$ and $\tilde{P}^\mu{}_\nu u_G^\nu = \tilde{P}^\mu{}_\nu n_\mu = 0$.

Using the velocity field $u_G^\mu$ we can define a Milne and $U(1)$ gauge invariant connection compatible with the Newton-Cartan data

$$
\begin{aligned}
\tilde{\Gamma}^\lambda_{\mu\nu} &= u_G^\lambda \partial_\nu n_\mu + \frac{1}{2}h^{\lambda\rho}\left(\partial_\mu \tilde{h}_{\nu\rho} + \partial_\nu \tilde{h}_{\mu\rho} - \partial_\rho \tilde{h}_{\mu\nu}\right) + \frac{1}{2}h^{\lambda\rho}\left(n_\mu \tilde{F}_{\nu\rho} + n_\nu \tilde{F}_{\mu\rho}\right), \\
\tilde{T}^\lambda_{\mu\nu} &= \tilde{\Gamma}^\lambda_{\mu\nu} - \tilde{\Gamma}^\lambda_{\nu\mu} = -u_G^\lambda F^{(n)}_{\mu\nu},
\end{aligned}
\tag{121}
$$

where we have defined the field strength

$$
\tilde{F}_{\mu\nu} = \partial_\mu \tilde{A}_\nu - \partial_\nu \tilde{A}_\mu.
\tag{122}
$$

The constitutive relations for Galilean fluids at leading order in a derivative expansion for the Milne-invariant stress-energy tensor $\mathcal{T}^{\mu\nu}$, energy current $\mathcal{E}^\mu$ and particle number current $J_c^\mu$ are

$$
\begin{aligned}
\mathcal{T}^{\mu\nu} &= P h^{\mu\nu} + \rho u_G^\mu u_G^\nu + \mathcal{O}(\partial), \\
\mathcal{E}^\mu &= \epsilon u_G^\mu + \mathcal{O}(\partial), \\
J_c^\mu &= \rho u_G^\mu + \mathcal{O}(\partial),
\end{aligned}
\tag{123}
$$

where $P$ is the pressure function, $\rho$ is the particle number density, and $\epsilon$ is the energy density. All these quantities are generic functions of the (Milne-invariant) temperature, $T$, and chemical potential, $\mu$, and satisfy the thermodynamic relations

$$
\epsilon = Ts + \mu\rho - P, \qquad d\epsilon = Tds + \mu d\rho,
\tag{124}
$$

where $s$ is the entropy density.

The equations of motion for Galilean fluids in a curved Newton-Cartan background are captured by the conservation of the stress-energy tensor and currents

$$
\begin{aligned}
(\tilde{\nabla}_\nu - \tilde{\mathcal{G}}_\nu)\mathcal{T}^{\mu\nu} &= -h^{\mu\rho}F^{(n)}_{\rho\nu}\mathcal{E}^\nu, \\
(\tilde{\nabla}_\mu - 2\tilde{\mathcal{G}}_\mu)\mathcal{E}^\mu &= -\frac{1}{2}\left(\tilde{h}_{\rho\mu}\mathcal{T}^{\mu\nu}\tilde{\nabla}_\nu u_G^\rho + \tilde{h}_{\rho\nu}\mathcal{T}^{\mu\nu}\tilde{\nabla}_\mu u_G^\rho\right), \\
(\tilde{\nabla}_\mu - \tilde{\mathcal{G}}_\mu)J_c^\mu &= 0.
\end{aligned}
\tag{125}
$$

Here,

$$
\tilde{\mathcal{G}}_\mu = \tilde{T}^\alpha{}_{\mu\alpha} = -F^{(n)}_{\mu\alpha}u_G^\alpha,
\tag{126}
$$

and $\tilde{\nabla}_\mu$ is the covariant derivative defined with the Milne invariant connection (121). See [27]. The leading order equations of motion for Galilean fluids can be obtained by inserting the expressions (123) in (125). After some massaging, one can rewrite these equations in the form $E_\mu = 0$, $E = 0$, and $E' = 0$, where

$$
\begin{aligned}
E_\mu &= \tilde{P}^\alpha_\mu \partial_\alpha P - \rho \tilde{F}_{\mu\alpha}u_G^\alpha + (P + \epsilon)F^{(n)}_{\mu\alpha}u_G^\alpha, \\
\tilde{E} &= -(\tilde{\nabla}_\mu - \tilde{\mathcal{G}}_\mu)(s u_G^\mu), \\
\tilde{E}' &= -(\tilde{\nabla}_\mu - \tilde{\mathcal{G}}_\mu)(\rho u_G^\mu).
\end{aligned}
\tag{127}
$$

By taking the flat spacetime limit

$$
n_\mu = (1,0), \qquad h^{\mu\nu} = \delta^{ij}\delta_i^\mu \delta_j^\nu, \qquad u_G^\mu = (1, v^i), \qquad A_\mu = 0,
\tag{128}
$$

where $v^i$ is the usual fluid velocity in Cartesian coordinates and $i = 1,\ldots d$ label the spatial coordinates, equations (127) reduce to the conventional Euler equation, continuity equation and entropy conservation.

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
