# Peer review of "Enstrophy from symmetry"

_SciPost Physics, doi:SciPost Phys. 12, 085 (2022)_

## Round 1 · Referee Report · Anonymous (Referee 1) · 2021-4-12

Report

This manuscript studies the enstrophy current for ideal fluids in 2+1 dimensions, and identifies the conserved quantity with a corresponding symmetry using an action formulation. The authors begin by enumerating a variety of candidates ($J_{(n)},J_h,J_G,J_{Gh},J_{conformal},J_c,J_H$), each of which are conserved. The focus of much of the paper is on $J$, which is a new U(1) generalization (10) which includes contributions from the chemical. The existence of (10) is then shown to only occur for specific equations of state, (29),(30). Both relativistic and Galilean fluids are considered.

I think the paper contains significant progress towards understanding enstrophy conservation (particularly (10) and the identification of the symmetry) and also meets all general acceptance criteria for SciPost. As such I recommend that the paper be published, though I would like to see the following points addressed:

  1. Given a number of competing definitions I think the notation could be clarified. For example, above (25a) it is claimed it is possible to generalize (10) to multiple U(1)s, and $\Omega$ appears in this definition. However the $\Omega$ appearing in (10) depends on a single chemical potential. Relatedly, the number of U(1) charges n (as in the argument: $\rho_1, \ldots, \rho_n$) matches the power of $\Omega$ appearing in this expression, it's not clear why there is a connection between these two numbers.

  2. The authors define F_{\mu\nu} as an external field strength (below (17)). For energy-momentum, this appears on the RHS as a source term which breaks energy-momentum conservation. Can F similarly source enstrophy in the charged fluid case? i.e. is it possible to relax the definition of enstrophy current adopted in this paper, and instead look for a current which is only conserved up to sourcing by F? Similar comments apply for the Galilean case with the Newton-Cartan geometric data h,n. If so, does this influence the allowed equations of state as in (29)/(30)?

  • validity: top
  • significance: good
  • originality: good
  • clarity: top
  • formatting: excellent
  • grammar: perfect

Author:  Natalia Pinzani-Fokeeva  on 2021-11-26  [id 1977]

(in reply to Report 1 on 2021-04-12)

We would like to thank the referee for carefully reading our manuscript. In what follows we reply to the two points raised and hope that the referee finds our reply appropriate.

  1. Inline equation above (25a) corrected: $n\rightarrow m$ to label multiple charges. Indeed, $\Omega$ will depend on multiple chemical potentials and the constraints arising from $\Omega_{\mu\nu}u^{\nu}=0$ will be modified. However, so long as $\Omega_{\mu\nu}u^{\nu}=0$ is satisfied, an enstrophy current exists in this case too.
  2. We have indeed analyzed this case too. When $F_{\mu\nu}u^{\nu}$ is present, eq (30) needs to be satisfied. Otherwise, the less stringent conditions (28) and (29) apply. For Galilean fluids, the absence of $F_{\mu\nu}$ does not lead to more relaxed conditions.

---

## Round 1 · Referee Report · Anonymous (Referee 2) · 2021-6-2

Report

The paper is well-written, well-thought, and makes interesting advances to our understanding of approximately conserved enstrophy currents. These explorations are particularly useful to further our understanding of turbulence in hydrodynamic systems starting from symmetry principles. However, while I believe in the technical soundness of the results, I find the manuscript lacking in substance and cannot recommend it for publication.

The authors identify a series of approximately conserved enstrophy currents for relativistic and Galilean hydrodynamics. They review the effective action formalism for ideal (non-dissipative) hydrodynamics and identify the accidental symmetry transformations responsible for the said approximate conservation. While these are interesting results, I believe that the paper itself lacks in weight. Most of the paper is either a review of enstrophy currents or of the effective action formalism for relativistic/Galilean hydrodynamics. The main results of the paper (derivation of the symmetry transformation for enstrophy) seem to be limited to eqs. (49)-(54) for the relativistic case, and the analogous Galilean statements. For this reason, I cannot rate the paper high in significance or originality.

Some interesting directions one could explore would be the physical interpretation of this symmetry. A fundamental ground-up motivation for this symmetry will also be helpful. For instance, the approximately conserved entropy current in relativistic hydrodynamics has been associated with the fundamental KMS transformation in, e.g., ref [5]. The authors should also note that the symmetry found in the paper is infinitesimal; it would have been nice to see some comments on the finite version of this symmetry.

Owing to the reasons mentioned above, I do not find the paper in accordance with the acceptance expectations of SciPost, and hence cannot recommend it for publication.
  • validity: high
  • significance: low
  • originality: low
  • clarity: good
  • formatting: excellent
  • grammar: excellent

Author:  Natalia Pinzani-Fokeeva  on 2021-11-26  [id 1976]

(in reply to Report 2 on 2021-06-02)

We beg to disagree with the referee's point of view. While we do review certain aspects of enstrophy and effective actions in this manuscript, our new results are more than the ones enumerated by the referee. In what follows we provide a distinction between what is new and what is not.

  • Enstrophy charge in non relativistic fluids is reviewed in Appendix A and a summary of existing results is collected at the beginning of Section 2, before 2.1 begins.
  • Our formulation in 2.1 is new as it provides the first instance where an enstrophy current has been identified for generic relativistic fluids, not necessarily neutral and conformal.
  • Our formulation in 2.2 is new in that it provides the first construction of enstrophy using a covariant formalism allowing us to show when a conserved enstrophy current exists in a curved background.
  • The effective action formalism for relativistic fluids is reviewed but it does not constitute a significant portion of our manuscript. In fact, it only appears at the beginning of Section 3.1 around equations 47, 48 and 51. The remainder of Section 3.1 is new as it provides a symmetry transformation for enstrophy as acknowledged by the referee.
  • The effective action formalism for nonrelativistic fluids in Section 3.2.1 is new.
  • The symmetry transformation for enstrophy in nonrelativistic fluids in Section 3.2.2 is also new as acknowledged by the referee.
  • Appendix B is a review of the Newton-Cartan formalism and Galilean covariant fluid dynamics.

---

## Round 1 · Referee Report · Anonymous (Referee 3) · 2021-6-25

Report

In this paper the authors focus their efforts on better understanding the enstrophy current. They begin by generalising known results to obtain the enstrophy current for relativistic, charged fluids as well as Galilean fluids, subject to certain constraints on the equations of state. In particular, they show that the corresponding enstrophy current is conserved in 2+1 dimensions given the vanishing of the vortex-stretching term. Then, the authors proceed to identify the associated symmetry giving rise to this conservation using an action formulation. The paper is very well-written and easy to understand and follow.

The importance of this work lies in the context of further understanding turbulent flows. As the authors review in the appendix, enstrophy conservation is key to the development of an inverse (as opposed to a direct) energy cascade in 2+1 dimensional incompressible non-relativistic flows. For the relativistic case, the situation is unclear. Furthermore, the dual manifestation of the conformal enstrophy current in AdS is particularly interesting and can teach us a lot about gravity through the AdS/CFT correspondence. For example, currently all known instabilities in AdS (i.e. super-radiance, weak turbulent instability) imply transfer of spectral weight to smaller scales. In this way, the turbulent instability in $D = 4$ bulk dimensions is particularly special and the key to understand this may lie with the relativistic enstrophy current and its geometric counterpart.

Given the above, I recommend this paper for publication in Sci.Post, subject to:
• expanding a little bit the introduction in order to highlight the importance of this work in the context of turbulent flows and the motivation for carrying out this computation.
• if possible, provide even a speculative physical interpretation of the symmetry (58). Also, explain why sign definite-ness of the divergence of the current is important.
• expanding the last paragraph of the Discussion section, where the authors discuss the implications of the enstrophy current conservation in the context of holography. Currently it doesn’t reflect the impact this might have on the gravity literature.
  • validity: top
  • significance: good
  • originality: good
  • clarity: top
  • formatting: perfect
  • grammar: perfect

Author:  Natalia Pinzani-Fokeeva  on 2021-11-26  [id 1975]

(in reply to Report 3 on 2021-06-25)

We would like to thank the referee for their comments. We address the minor suggested request as follows - We added a few sentences at the end of the 4th paragraph on Page 2 to address points 1 and 2 - The beginning of the paragraph that contains the new equations Eqs 58 and 59 has been added to address point 2 - The last paragraph at the end of the discussion on page 17 expanded to address point 3

Moreover, we have made the following changes: - Typo corrected on page 5: g=s/rho -> g=rho/s

We hope the referee finds the adjustments appropriate.

---

## Round 2 · Referee Report · Anonymous (Referee 3) · 2021-12-13

Report

I am happy with the changes that the authors have made. I think this paper meets the criteria of the journal and I thus recommend the paper for publication.

---

## Round 2 · Referee Report · Anonymous (Referee 1) · 2021-12-26

Report

I am satisfied that the authors have addressed my concerns and I am happy to recommend publication in SciPost.

---

## Round 2 · Referee Report · Anonymous (Referee 2) · 2022-2-13

Report

I thank the authors for clarifying the significance of their paper. I reserve my opinion about the substance and impact of the work but, since I have no technical disagreement with the authors, I am happy to leave the decision of publication with the editor.

---

## Round 2 · List of Changes

• We added a few sentences at the end of the 4th paragraph on Page 2 to address the comments of referee 3
  • Typo corrected on page 5: g=s/rho -> g=rho/s
  • Eqs 58 and 59 were added to address the comments of referee 3
  • The last paragraph at the end of the discussion on page 17 expanded to address the comments of referee 3
  • Inline equation above 25a corrected: n->m to label multiple charges. Indeed, Omega will depend on multiple chemical potentials and the constraints arising from Omega_mnu^n=0 will be modified. However, so long as Omega_mnu^n=0 is satisfied, an enstrophy current exists in this case too. This change addresses the comments of referee 1.

---

## Editorial Decision

published